# Iron Supplementation Increases Tumor Burden and Alters Protein Expression in a Mouse Model of Human Intestinal Cancer

**DOI:** 10.3390/nu16091316

**Published:** 2024-04-27

**Authors:** Ian X. Swain, Adam M. Kresak

**Affiliations:** Department of Pathology, School of Medicine, Case Western Reserve University, 2103 Cornell Road, Cleveland, OH 44106, USA; adam.kresak@uhhospitals.org

**Keywords:** iron supplementation, micronutrient, intestinal cancer, protein expression, tumor, mouse model

## Abstract

Iron supplements are widely consumed. However, excess iron may accelerate intestinal tumorigenesis. To determine the effect of excess iron on intestinal tumor burden and protein expression changes between tumor and normal tissues, *Apc^Min/+^* mice were fed control (adequate) and excess iron (45 and 450 mg iron/kg diet, respectively; *n =* 9/group) for 10 wk. Tumor burden was measured, and two-dimensional fluorescence difference gel electrophoresis was used to identify differentially expressed proteins in tumor and normal intestinal tissues. There was a significant increase (78.3%; *p* ≤ 0.05) in intestinal tumor burden (mm^2^/cm) with excess iron at wk 10. Of 980 analyzed protein spots, 69 differentially expressed (*p* ≤ 0.05) protein isoforms were identified, representing 55 genes. Of the isoforms, 56 differed (*p* ≤ 0.05) between tumor vs. normal tissues from the adequate iron group and 23 differed (*p* ≤ 0.05) between tumors from the adequate vs. excess iron. Differentially expressed proteins include those involved in cell integrity and adaptive response to reactive oxygen species (including, by gene ID: ANPEP, DPP7, ITGB1, PSMA1 HSPA5). Biochemical pathway analysis found that iron supplementation modulated four highly significant (*p* ≤ 0.05) functional networks. These findings enhance our understanding of interplay between dietary iron and intestinal tumorigenesis and may help develop more specific dietary guidelines regarding trace element intake.

## 1. Introduction

Despite the prevalence of early detection methods, colon cancer remains the second leading cause of cancer-related death [1]. Iron supplements are widely consumed in the U.S. [2]. However, most supplemental (non-heme) iron remains unabsorbed [3]. Although micronutrients such as iron play a vital role in tissue growth and cellular homeostasis, excess iron may promote infection, neoplasia, cardiomyopathy, and arthropathy and exacerbate endocrine and neurodegenerative disorders [4,5,6,7,8]. Free intraluminal iron may also act as a prooxidant-inflammatory agent. Unabsorbed dietary iron becomes available for participation in a combination of Haber–Weiss and Fenton-type reactions that generate hydrogen peroxide and hydroxyl radicals at the mucosal surface [9,10]. Iron is required for cell proliferation, yet in excess may accelerate growth of mutated cells [11]. Therefore, unabsorbed intraluminal iron may plausibly act in the initiation of carcinogenesis by causing DNA damage and at the promotion stage by stimulating polyp growth or the accumulation of additional mutations.

The epidemiologic evidence supporting a link between diet and cancer of the intestinal tract, especially the colon, is strong [12]. Understanding how individual components of the diet modify cancer risk continues to be an important topic [13]. Evidence from previous studies continues to support the concept that dietary iron plays an important role in determining risk for intestinal cancer. The hypothesis that high iron intake may be related to increased risk for developing intestinal tumors in humans was based on epidemiological evidence showing a clear association between higher meat intake and cancer of the colon and rectum [14,15]. Data from a National Health and Nutrition Examination Study (NHANES), a prospective study of over 10,000 people in the northeastern U.S., showed that men who were diagnosed with colon cancer had significantly greater transferrin saturation on recruitment as compared with controls [16]. Likewise, other studies found a positive associated between cancer risk in a variety of tissues and transferrin saturation [17,18,19,20,21,22,23,24]. A study by Freudenheim et al. [22] revealed that high iron intake was associated with a significantly increased risk of intestinal cancer. Data from another NHANES study not only demonstrated a link between high body iron stores, but, in particular, showed a significant association between high dietary iron and colorectal cancer [21]. More recently, a World Cancer Research Fund report also described evidence for increased colorectal cancer risk with excess dietary iron intake [25].

Familial adenomatous polyposis (FAP) is a human colon cancer predisposition syndrome in which a single mutated allele of the adenomatous polyposis coli (APC) gene is inherited. Most colon cancers (~80%) have mutations that develop somatically in the APC gene [26]. *Apc^Min/+^* mice are a murine model of human FAP, possessing a germline mutation in the Apc tumor-suppressor gene [26,27]. Since at least one allele of APC is mutated in 80% of sporadic colorectal cancers in humans, the *Apc^Min/+^* mouse is a relevant model for studying intestinal tumorigenesis and tumor growth [28,29]. Understanding how excess dietary iron influences tumor growth and protein expression changes in tumor and normal intestinal tissues in *Apc^Min/+^* mice would provide insight into how supplemental dietary iron may influence human intestinal cancer.

Nutritional proteomics in cancer prevention is a rapidly progressing area of study. The importance of studying cancer-related proteomics in tissues in vivo has been emphasized [30] and underscored during a National Cancer Institute meeting (Nutritional Genomics and Proteomics in Cancer Prevention Conference, National Institutes of Health (NIH), MD, USA) [31]. Studies from breast [32], prostate [33], ovarian [34], and colon [35] cancer typically identify hundreds of differentially expressed proteins, some over- and others under-expressed. However, these differentially expressed proteins may or may not be relevant to carcinogenesis. Determining whether and how excess dietary iron influences the proteome of tumor and normal intestinal tissue in *Apc^Min/+^* mice, combined with morphological assessment of tumors, may help explain the mechanism(s) of iron-mediated intestinal tumorigenesis.

The purpose of this study is to determine the effect of iron supplementation (excess dietary iron) on intestinal tumor burden and proteomic changes between tumor and non-tumor tissues from intestines of *Apc^Min/+^* mice fed adequate and excess iron. The aim is to obtain data to help develop more specific dietary guidelines regarding human iron intake and intestinal cancer, aimed at slowing the progression of benign tumors to carcinomas.

## 2. Materials and Methods

### 2.1. Animals and Study Design

Eighteen male C57BL/6J *Apc^Min/+^* mice (Jackson laboratories, Bar Harbor, ME, USA) were obtained at 5 weeks of age. Upon arrival, mice were acclimated for 24 h before being randomly assigned to treatment groups (*n* = 9/group). Dietary treatment lasted for 10 weeks, at which point mice were sacrificed, tumor and non-tumor intestinal tissues collected, and analyses performed. Mice were housed individually in microisolator cages in a room controlled for temperature (21 ± 1 °C), humidity, and light (12 h light:dark cycle). Mice consumed water and food ad libitum. Food intake and overall health were monitored once every three days, as fresh diet was provided and uneaten/spilled food was measured on the same schedule. Body weights were monitored weekly. All animal procedures followed the Institutional Animal Care and Use Committee procedures at Case Western Reserve University (CWRU), in accordance with the NIH guidelines.

### 2.2. Diet

Iron (ferrous sulfate monohydrate: FeSO_4_·H_2_O; 32.7% iron (Fe) *w*/*w*; Crown Technologies, Inc., Indianapolis, IN, USA) was incorporated into an AIN-93M[M] diet [36] (Cat.#TD.99397; Harlan Teklad, Madison, WI, USA) modified to omit the iron source (no ferric citrate). Diet constituents are shown in Table 1. Diets were prepared from the modified preparation (AIN-93M[M]), each with a different iron content. Mice were randomly assigned to receive either of the two diets: control (adequate) iron at 45 mg Fe/kg diet or excess iron at 450 mg Fe/kg diet (*n* = 9/group). Atomic absorption spectrophotometry was used to confirm iron concentrations of the diets, as previously described [37]. The concentration (mean ± SEM) of iron assayed in the 45 and 450 mg Fe/kg diets, performed in triplicate, were 44.8 ± 0.7 mg and 446.9 ± 6.2 mg iron/kg diet, respectively. Diets were stored at −20 °C. Fresh diet was given to mice every third day, with uneaten pellets discarded after weighing the unused portion and any spilled diet. Ferrous sulfate monohydrate is a highly bioavailable form of non-heme iron and is often used as an iron supplement for humans [38]. The excess iron murine diet used in this study represents a level of human iron intake associated with daily use of iron supplements and regular consumption of highly fortified foods by humans [39,40].

### 2.3. Tissue Collection

At the end of the 10th week of treatment, mice were fasted overnight before euthanasia and tissue collection. Mice were sacrificed by cardiac puncture following intraperitoneal administration of sodium pentobarbital (50 mg/kg body weight). Intestines were removed from duodenum through the colon, excluding the caecum, rinsed with ice-cold phosphate-buffered saline, and weighed before opening longitudinally dissecting scissors. Collection of normal and tumor intestinal tissues was performed using a micro-dissection scalpel under 10× magnification using a Leica M125 C stereo-microscope (Leica Microsystems, Buffalo Grove, IL, USA), then placing the tissue in ice-cold 2.15 mL buffer (pH 7.0, 0.05 mol/L potassium phosphate). The samples were then flash frozen in liquid nitrogen until use.

### 2.4. Tumor Burden

Tumors were identified and measured in intestinal spread flat, mucosal surface up, on bibulous paper, kept cold during evaluation in a standard cold room, and using an ice tray below with a layer of plastic wrap in between. Tumors were counted at 10× magnification using a Leica DM6000 wide-field microscope with multidimensional workstation, with LAS X software version 1.4.6 (Leica Microsystems, USA). The smallest distinguishable tumors were ~0.5 mm in diameter. The method for scoring is similar to that described previously [41]. Microscopic analysis included measurement of tumor cross-diameters in micrometer units to enable calculation of tumor burden (mm^2^/cm). To confirm tumors (adenomas or carcinoma, and not lymphoid aggregates), tissue samples were randomly selected for confirmational analysis by the CWRU Histology Core Center.

### 2.5. Proteomic Analysis—2D-DIGE (2-Dimensional Fluorescence Difference Gel Electrophoresis)

Next, 2D-DIGE experiments for each sample, in triplicate, were performed using protein extracts from normal and tumor intestinal tissues of *Apc^Min/+^* mice fed adequate (control) and excess iron. Solubilization of cells was performed in a lysis buffer containing 2M thiourea, 7M Urea, 4% CHAPS, and 25 mM Tris, pH 8.5, with lysates centrifuged at 12,000 rpm for 4 min, using a Thermo Scientific SL1R Plus Centrifuge (ThermoFisher Scientific, Asheville, NC, USA). Protein concentration of the supernatant used was determined by ion-exchange analyses (PK7300 Automated Microplate System; Beckman Coulter, Inc., Brea, CA, USA). For each tissue sample, proteins (50 μg) were labeled with 400 pmol of Cy3 or Cy5 N-hydroxysuccinimidyl ester dyes for 30 min (Amersham Biosciences, Piscataway, NJ, USA). For each gel set, an additional gel was run where samples one and two were reverse labeled. Labeling reaction quenching was conducted with 0.2 mM lysine and a matching volume of buffer containing 2M thiourea, 7M Urea, 4% CHAPS, 2% DTT, and 2.0% Pharmalyte (Amersham Biosciences). An internal standard was obtained by pooling an aliquot of all samples and labeled with Cy2 dye; hence, the three samples in any gel comparison (i.e., tissue sample 1, tissue sample 2, and internal standard). Rehydration of the IPG strip (pH 3–10, 24 cm) with protein samples was performed in buffer containing 8M Urea, 4% CHAPS, bromophenol blue, 1% Phamalyte, and 2 mg/mL DTT using the IPGphor IEF system (Amersham Biosciences), at 20 °C for 12 hr followed by isoelectric focusing at 120 kVh at 20 °C. Thereafter, gel strips were equilibrated for 2 × 15 min, while gently shaking in equilibration solution (50 mM Tris-Cl buffer, 6M urea, 30% glycerol, 2% SDS, and bromophenol blue). DTT (1% *w*/*v*) was added to the first, and iodoacetamide (2.5% *w*/*v*) at the second equilibration step. SDS polyacrylamide gel electrophoresis in the second dimension was carried out using homogenous 12.5% polyacrylamide gels and the Ettan Dalt II large vertical system (Amersham Biosciences). Labeled proteins were visualized using the Typhoon 9410 imager (Amersham Biosciences). The fluorescence images were individually captured using excitation/emission wavelengths of 488/520 nm, 532/580 nm, and 633/670 nm for Cy2, Cy3, and Cy5, respectively. DeCyder software v.6.5 (Amersham Biosciences) was used for image analysis, including spot detection, background subtraction, matching, and normalization. The average values of the reverse-labeled samples were used for differential analysis, which was performed using R-after log-transformation. Differentially expressed proteins were determined based upon one-way ANOVA followed by Tukey’s multiple comparison test. Prior to protein excision, the gel was post-stained using deep purple staining protocol as described by the manufacturer (Amersham Biosciences), scanned using a 457 nm laser and an emission filter of 610 nm BP (band pass) 30, and used as preparative gel for downstream protein identification. Proteins of interest were then excised using an Ettan Spot Picker (Amersham Biosciences), then digested with sequencing-grade trypsin (Promega Corp., Madison, WI, USA). Tandem mass spectra of digested peptides were acquired using Fourier Transform LTQ mass spectrometer (FT-LTQ, Thermo Electron Corp., Bremen, Germany) equipped with HPLC system and LTQ mass spectrometer (Thermo Electron Corp.) with Ettan MDLC system (GE Healthcare, Chicago, IL, USA). Protein digests were loaded onto a trapping column (C18, PepMap100, 300 μm × 5 mm, 5 μm particle size, 100 Ǻ, Dionex, Sunnyvale, CA, USA) to pre-concentrate, then separated with a reverse phase column (C18, 75 μm × 150 mm, 3 μm, 100 Ǻ, Dionex) using mobile phase A (0.1% formic acid) and B (84% acetonitrile, 0.1% formic acid) at a linear gradient of 2% per min. Peptides were then infused at a flow rate of 300 nL/min and at a voltage of 1.8 kV, with MS/MS spectra obtained in positive ion mode. Full scans were recorded in the FT analyzer (resolution R = 100,000) followed by MS/MS in the LTQ analyzer (FT-LTQ, Thermo Electron Corp., Bremen, Germany). Protein identification was determined by searching rodent NCBInr and NCBIprot databases using MASCOT v.2.1.03 and v.2.4 and as previously described [42,43]. Searches were performed with criteria including carbamidomethylation of cysteine, with partial oxidation of methionine, with one missed cleavage allowed, and with mass tolerance of 15 ppm and of 0.8 Da for MS and MS/MS, respectively, for FT-LTQ and with mass tolerance of 1.5 Da and of 0.8 Da for MS and MS/MS, respectively, for LTQ. Protein identification was determined based on protein “hits” with at least two matched peptides and a peptide ion score exceeding forty-five.

### 2.6. Pathway Mapping

Ingenuity Pathways Analysis (IPA; Qiagen, Germantown, MD, USA) was performed to reveal biologically relevant protein expression/gene networks related to cancer (Pathway Studio 7, Ariadne Genomics, Rockville, MD, USA).

### 2.7. Statistical Analyses

Power analyses were performed based on previous published data [35]. For tumor burden measurements, treatment means ± SEM, medians, one- and two-way ANOVAs, tests for normal distribution of data and nonparametric procedures, were calculated using the statistical package SAS (SAS Version 10.2, SAS Institute, Cary, NC, USA). For proteomic determinations, ANOVA and fold-change differences were used to determine differential significance; proteins presented are those for *p* ≤ 0.05 from analysis of variance (ANOVA), followed by Tukey’s means separation test with fold-change. Data were examined for normality of distribution and homogeneity among differences. Differences were considered significant if *p* ≤ 0.05. Illustration of data and results was performed using GraphPad Prism (Software version 10.2; GraphPad, Boston, MA, USA).

## 3. Results

### 3.1. Animal Weight and Food Intake

The rate of animal weight gain and final weights did not significantly differ (*p* ≤ 0.05) between the diet groups; from wks 5 to 10, weight gain (mean ± SEM) for the adequate and excess iron diet groups were 1.8 ± 0.5 and 1.9 ± 0.6 g/wk, respectively, and at wk 10 final animal weights for the adequate and excess iron diet groups were 26 ± 1.1 and 25.9 ± 1.4 g, respectively. Food intake also did not significantly differ (*p* ≤ 0.05) between the treatment groups throughout the study; at wk 5, food intake (mean ± SEM) for the adequate and excess iron diet groups was 1.8 ± 0.6 and 1.7 ± 0.8 g/d, respectively, and at wk 10 food intake for the adequate and excess iron diet groups was 2.4 ± 0.7 and 2.3 ± 0.9 g/d, respectively.

### 3.2. Tumor Identification, Measurements, and Burden Calculation

Intestinal tumors (full-length intestines—proximal duodenum through colon, excluding caecum) were identified and confirmed via histological analysis, and cross-section/diameters were measured (Figure 1A shows image of group of tumors found, illustrating bi-dimensional measurements in micrometers). Tumor counts revealed that tumor number (mean ± SEM) did not differ significantly differ (*p* ≤ 0.05) between the adequate iron and excess iron diet groups, at 34 ± 5 and 37 ± 6, respectively. However, there was a significant increase (78.3%; *p* ≤ 0.05) in tumor burden (mm^2^/cm) with excess as compared to adequate iron at wk 10 (2.3 ± 0.6 vs. 4.1 ± 0.9, respectively; see Figure 1B and Appendix A).

### 3.3. Comparative Proteomic Analysis of Tumor and Non-Tumor Intestinal Tissue

Proteomic analysis of tumor and non-tumor intestinal tissues revealed differently expressed proteins between normal and tumor intestinal tissues, and between tumor-to-tumor intestinal tissue at the two different levels of dietary iron intake. Figure 2A–C show a sample 2D-DIGE gel overlays, sequence subset gels, and a 2D (deep purple-labeled) protein spot map of tumor tissue comparisons from the adequate vs. excess iron groups. Figure 3A,B show those tumor tissue sample images of 2D-DIGE Cy3 and Cy5 labeled proteins from intestinal tumor tissues from the adequate vs. excess iron groups, respectively, to illustrate differential protein expression. Figure 4 shows the associated 2D map, protein spot coordinates associated with the Cy-labeled sample coordinates, and pH gradients with approximate molecular mass ranges. Appendix A shows hierarchical clustering of differentially expressed protein spots in tumor as compared to adjacent non-tumor intestinal tissue when comparing the diets.

Of 980 analyzed protein spots, we identified 69 differentially expressed (*p* ≤ 0.05) protein isoforms overall, representing 55 genes, when tumor tissue was compared to normal tissue from the adequate iron diet group and when tumor tissues between the adequate and excess iron groups were compared (Table 2). Among these 69 protein isoforms, 56 significantly changed (*p* ≤ 0.05) in tumor as compared to normal tissue in the adequate iron group (8 decreased, 48 increased) and 23 significantly changed (*p* ≤ 0.05) between intestinal tumors from the adequate as compared to the high iron group (7 decreased, 16 increased); some of the same proteins exhibited differential expression in both comparisons but the degree of fold-change in expression was different.

Some of these differentially expressed proteins are involved in modulation of reactive oxygen species (gene name: SOD1, SOD2, PRDX1, ITGB1, and ALB) and others in protein degradation (ANPEP, DPP7, PSMA1, PSMA2, PSMA3, SERPINB1, HSPA5, HSPA9, PSMA4, PEPD). Of particular interest were differently expressed proteins in tumors from mice fed adequate as compared to excess iron, which included those involved in cell integrity (ANPEP: alanyl (membrane) aminopeptidase; DPP7: dipeptidyl-peptidase; PSMA1: proteosome subunit; and SERPINB1: serine peptidase inhibitor) and adaptive response to reactive oxygen species (SOD1: superoxide dismutase); SOD2; and PRDX1: peroxiredoxin), possibly related to the increased cell turnover and growth of tumor cells and concurrent adaptation of tumors to increased intraluminal dietary iron. Figure 5A,B show key protein fold-change expression differences.

### 3.4. Pathway Analyses

Using IPA, we found four highly significant functional networks in tumor compared to normal tissue and two in tumor tissues from the adequate as compared to high iron (Figure 6A,B). Notably, the top network in both cases was associated with functions implicated in cancer and cell death. Related biological pathways (functions) modulated by excess dietary iron, in addition to cancer, cell cycle, and cell death, include gastrointestinal (GI) disease, inflammatory disease and free-radical scavenging, DNA replication, recombination, and repair, and protein folding and trafficking (Appendix A).

## 4. Discussion

In this study, excess dietary iron was associated with changes in tumor burden and expression of proteins in intestinal tumors. Our findings concur with other studies that have shown higher dietary iron intake is associated with increased levels of epithelial cell proliferation and tumor development and growth [44]. This study’s data are also in agreement with earlier research showing that increased dietary iron intake is associated with an altered inflammatory response, chronic gut inflammation, and hyperproliferation of the intestinal epithelium [45]. Prior studies that have identified associations between increased intestinal epithelial intracellular iron uptake and colorectal tumorigenesis have also identified an oxidative intracellular imbalance [46,47,48]. However, our findings provide evidence of differential expression of specific protein isoforms involved in modulating oxidative response of tumor cells with higher (excess) dietary iron.

Our data show that in tumors from the adequate iron diet group, as compared to non-tumor tissue, there were significant changes (*p* ≤ 0.05) in expression of proteins involved in modulation of reactive oxygen species (ROS) (SOD1, SOD2, PRDX1, ITGB1, ALB, and VDAC2; all protein abbreviations and identities are defined and described in Table 2). With excess dietary iron, these protein expression changes in tumors are even more pronounced—increased when comparing tumor tissue between the adequate and excess diet groups. SOD1 is known canonically as an antioxidant enzyme; increased levels of SOD1 decrease intracellular ROS levels [49]. It has been shown that loss of SOD1 inhibits cell proliferation and survival. However, there is no evidence for tumor growth dependent on SOD1 enzymatic activity, while there is for increased cell signaling and Paneth cell differentiation in SOD1 deficient cells [50]. Overexpression of SOD2 induces a radioactivity-sensitizing effect on existing tumor cells and a protecting effect on normal cells [51]. Compounds targeting ROS-manipulating protein PRDX1 revealed that decreased levels of PRDX1 suppressed colorectal cancer cell proliferation even under adequate dietary iron concentrations [52]. Increased PRDX1 levels have also been observed in inflammatory models, and silenced PRDX1 expression inhibited these inflammatory responses [53]. ITGB1 is known to have tumorigenic effects, and inhibition of ITGB1 inhibits cancer cell proliferation [54]. It has also been observed that some tumor-suppressor transcription factors are negatively correlated with ITGB1 expression, further providing evidence of ITBG1′s tumor-promoting effect [55]. The role of albumin (ALB) has not been deeply explored in the context of altered dietary iron and intestinal cancer. It has been observed that albumin, when administered as fusion protein ALB-IL2, demonstrates a T cell-mediated anticancer effect in colon tumors [56]. This could be related to albumin’s localization and accumulation properties and its role as a biological transporter. VDAC2, a voltage-dependent anion channel, enhances the production of reactive oxygen species upon binding with the ferroptosis inducer erastin. It has been demonstrated that preventing the degradation of VDAC2 and VDCA3 supports cell sensitivity to erastin through this process [57]. Tumor spheroids exhibit relatively lower expression levels of VDAC2 compared to normal tissues [58]. Notably and unexpectedly, our results highlight an increased level of VDAC2 expression (an increased fold-change of 10.86 (*p* ≤ 0.05) when comparing tumor tissues between the excess iron vs. adequate iron groups), most likely in cells undergoing ferroptosis. Ferroptosis is a type of programmed cell death that is dependent on iron and characterized by the accumulation of lipid peroxides but is genetically and biochemically distinct from other programmed types of cell death such as apoptosis [59,60]. Therefore, when our protein expression and tumor burden data are combined, we found a strong pattern of association between tumor growth and expression of proteins involved in adaptive response to oxidative balance; possibly reflecting that as tumors utilize excess dietary iron for accelerated growth, they experience a need to adapt to the increased pro-oxidative micro-environment created by excess intraluminal iron.

Within tumor tissue in the adequate iron group, and more so in tumors from the excess iron diet group, we found there was also significantly greater (*p* ≤ 0.05) expression of proteins involved in protein degradation (ANPEP, DPP7, ITGB1, PSMA1, PSMA2, PSMA3, and SERPINB1). ANPEP has not been as widely studied, especially in the context of intestinal cancer development with varied iron intake, but it has been observed that ANP/ANPEP is downregulated in tumorigenic prostate cell lines [61,62]. Likewise, DPP7′s potential role in intestinal cancer has not been widely noted. Yet, there is prognostic significance to DPP7; it is shown that higher DPP7 expression is correlated to a higher patient survival rate in the presence of colon adenocarcinomas, and that DPP7 is significantly under-expressed in tumor tissue [63]. Further, p53-mediated inhibition of DPP4 (in the dipeptidyl peptidase family) has been shown to promote cell survival during adequate iron intake [64]. Protease PSMA1, involved in intracellular protein degradation, is known to be tumorigenic. It has been demonstrated in gastric and lung cancer tissues that PSMA1 is upregulated [65,66], and PSMA1 is a biomarker of colon cancer [67]. Colorectal cancer cells, when treated with an anti-cancer compound, exhibited down-regulation of PSMA1 [68]. Additionally, PSMA2 is a protease with established potential as a biomarker for ovarian cancers [69,70]. In the context of colorectal cancer, PSMA2 enhanced proliferation, migration, and invasion of tumor cells and showed increased expression in stages 1–4; when inhibited, the rate of tumor cell proliferation, which is dependent on iron availability, was dampened [71]. Similarly to PSMA2, protease PSMA3 is found to be highly expressed and significantly hindered proliferation, migration, and invasion of tumor cells in colorectal cancer [72]. Proteinase inhibitor SERPINB1 has not been widely explored in relation to intestinal or colorectal cancer, but proteomics results indicate that SERPINB1 is downregulated in skin and prostate tumor tissues [73,74]. Since SERPINB1 assists in protease inhibition, dysregulation may allow cellular protein degradation to persist unchecked, as seen in other cancer development pathways involving iron as a requirement for cellular growth.

Excess dietary iron in tumor tissues was also associated with significant changes (generally, increases; *p* ≤ 0.05) in proteins involved in modulating cell protein integrity (HSPA5, HSPA9, ITGB1, PSMA4, DPP7, and PEPD), cell mobility and growth (CAPZB), and immunologic factors (FCGBP). HSPA5 acts as a chaperone protein to regulate protein folding. It has been shown to contribute to tumorigenesis and anti-apoptotic properties, evidenced by significantly reduced survival and increased apoptosis of cancer cell lines with inhibition of HSPA5 [75]. HSPA9 is a protein with a role in cell proliferation and metabolism, also acting as a chaperone protein. HSPA9 was found to be a key gene in determining clinical outcomes in resected colorectal cancer patients, hinting at differing expression levels in tumor stem cells compared to normal stem cells in the digestive tract [76]. It has been noted that HSPA proteins are ferroptosis inhibitors, which are influenced by iron, and that they contribute to the shutdown of cell death, further promoting tumor growth [64]. Hence, one of the key findings of this study may be that tumor cells adapt to excess dietary iron by expressing proteins that enable them to simultaneously utilize greater iron to accelerate growth, while modulating/inhibiting pathways that would limit such growth. PSMA4, another proteasome, shares a similar fate as PSMA1, being that it is downregulated in cancer cells when those cells are treated with an anti-cancer compound [68]. PEPD is a peptidase that has been shown to inhibit tumor signaling in colorectal cancer through promoting epidermal growth factor receptor inhibition [77]. As a peptidase, the degradation of growth factor receptors, key players for development of tumorigenic cells, is useful for inhibiting cancer growth, especially because some cancers may be resistant to inhibitors. CAPZB encodes a subunit of an actin binding cap protein that regulates actin assemblage in muscle cells. The increased CAPZB fold-change we observed (+10.86; *p* ≤ 0.05) when comparing tumor tissues between the excess and adequate iron groups is in agreement with other studies that have investigated expression of CAPZB in epithelial sarcoma cell lines is associated with increased cell mobility and growth, as CAPZB knockouts prevent cell migration [78]. Other studies have shown that when CAPZB is targeted, its ability to facilitate tumor metastasis is suppressed [79]. Furthermore, high CAPZB expression can be expected in tumorigenic tissues as it mediates increased invasion and migration tendencies, possibly even outside of muscle tissue. The effect of excess dietary iron on immunologic response and activity of IgGFc-binding protein (FCGBP) in intestinal epithelial cells has not been thoroughly studied. In this study, we observed increased FCBGP expression (fold-change of +15.21; *p* ≤ 0.05) when comparing tumors to normal tissue in the adequate iron group. In the intestine, FCGBP complexes promote cell migration by facilitating binding to trefoil factors [80,81]. FCGBP has been shown to be upregulated in glioma tissues and to be associated with increased immune cell invasion [82]. Further investigation is warranted to understand FCGBP’s role in inflammation and the immune response due to a high-iron environment. The identification of the need to better elucidate the interplay between expression of proteins that enable tumor growth, offer oxidative adaptation, and mediate immunologic response while maintaining tumor cell integrity via inhibitory action is another key finding of this study.

One limitation of this study may be that other concentrations of dietary iron in between the adequate (control) iron diet group and the excess iron diet group were not included due to availability of resources. However, the number of animals per group in our study was calculated and utilized in order to provide greater statistical power; hence, the reason for our statistically strong differential expression findings between tumor and normal tissues. Also, there exist other types of animal models of human intestinal cancer, such as *Apc^1638N/+^* mice. Both *Apc^1638N/+^* mice and the mouse model used in this study (*Apc^Min/+^*) possess mutations of the Apc locus that inactivate the gene. We used *Apc^Min/+^* mice because there is more extensive literature describing intestinal tumorigenesis in these mice. Further, more is known about the effect of micronutrients on cancer development in the *Apc^Min/+^* mouse model, providing more opportunities for comparing our findings to the literature.

Network analysis found Myc-, p53-, and TGFB-related networks in tumor as compared to normal tissue in mice fed adequate iron and Myc- and p53-related networks in tumor tissues from mice fed excess compared to adequate iron. Remarkably, the top network in both cases was associated with functions implicated in cancer and cell death. Iron is required for DNA synthesis and proliferation, and it reaches cellular targets that influence the cell cycle and cell death [83]. Iron-related metabolism and signaling pathways are also showing promise as targets in cancer treatment. This includes not only indirect adjustments of net iron intake, but also regulation of ferroptosis, removal of serum iron, and the mediation of biosignaling and oxidative stress pathways occurring in various cellular locations. Interestingly, the metastasis suppressor N-Myc downstream-regulated gene-1 (NDRG1) is activated by iron chelators [84]. Also, transferrin receptor 1, required for iron importation from transferrin into cells via endocytosis, has been identified as a downstream target of the Myc network and is overexpressed in cancer tissue, contributing to higher levels of cellular iron intake [85]. Depleted iron levels have been found to promote an anti-tumor status in multiple ways, including regulation of p53, a cell cycle regulator [86,87]. p53 has also been shown to promote both cell survival and death through ferroptosis-mediated approaches, but the exact interaction between p53 and ferroptosis and perhaps the existence of a molecular switch to apoptosis are not clear. It is simply known that p53 is a major regulator and that iron is a key participant in its regulation pathway.

## 5. Conclusions

These findings enhance our understanding of the interplay between dietary iron and intestinal tumor development. Notable changes in protein expression in tumor and normal intestinal tissues also give insight into metabolic, biochemical, and intracellular pathways associated with dietary iron and intestinal cancer. Combining a comparative proteomic analysis of these tissues with tumor burden data increases our understanding of differentially expressed proteins and how they may be relevant to cancer progression. Data from this study may also help develop more specific dietary guidelines regarding iron and cancer.

## Figures and Tables

**Figure 1 nutrients-16-01316-f001:**
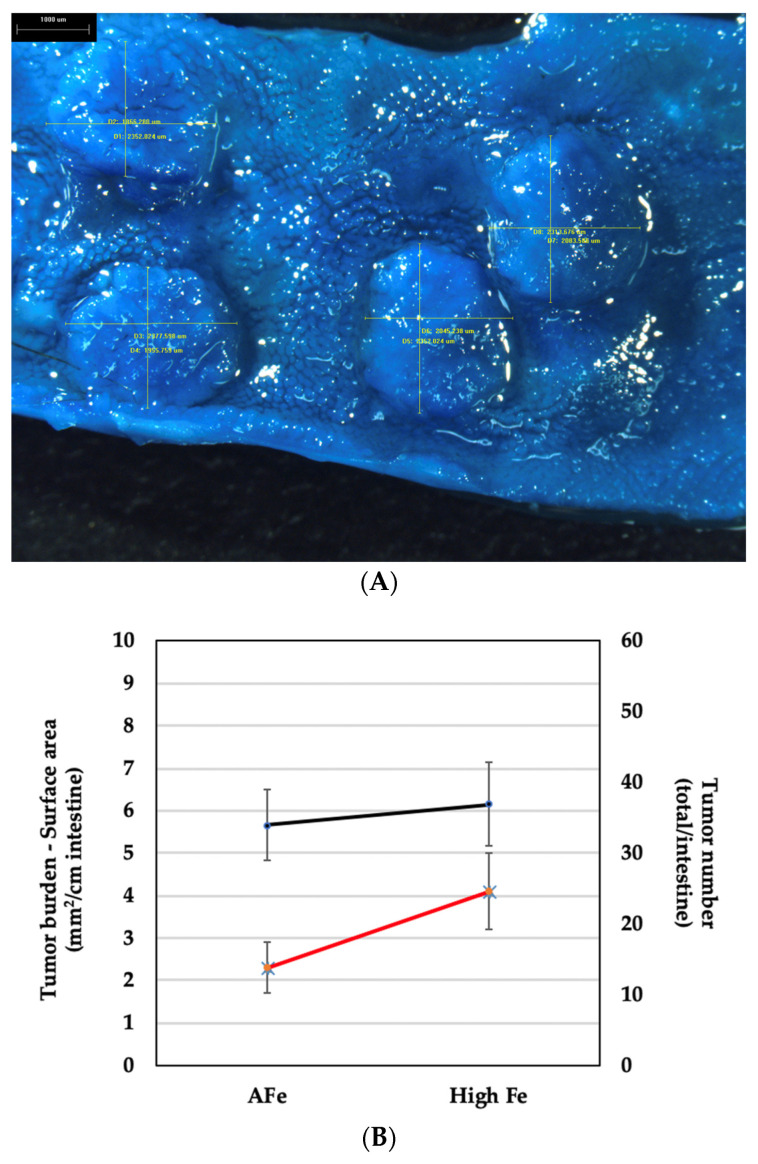
(**A**) Intestinal segment opened longitudinally and stained with methylene blue to show tumors. Bi-dimensional measurements are also shown (“um” = μm); cross-section (diameter) measurements (lines) overlay each tumor. Normal villi can be seen surrounding tumors. Bar in upper left corner = 1000 μm (1 mm). (**B**) Tumor number (total/intestines) and tumor burden (mm^2^/cm) in intestines of mice fed adequate (AFe) and excess (High Fe) iron, at wk 10. (Black line represents tumor number and red line tumor burden; values are mean ± SEM for each treatment group).

**Figure 2 nutrients-16-01316-f002:**
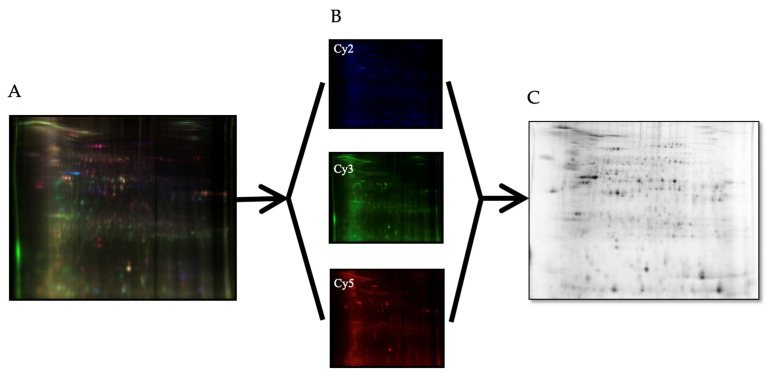
(**A**) Overlay of 2D-DIGE fluorescence gel images, (**B**) individually captured fluorescence gel images using excitation/emission wavelengths of 488/520 nm, 532/580 nm, and 633/670 nm for Cy2 (internal standard), Cy3 (tumor tissue; adequate iron diet), and Cy5 (tumor tissue; excess iron diet), respectively, and (**C**) the accompanying 2D deep purple-labeled map of the intestinal tissue proteome, with spots of interest.

**Figure 3 nutrients-16-01316-f003:**
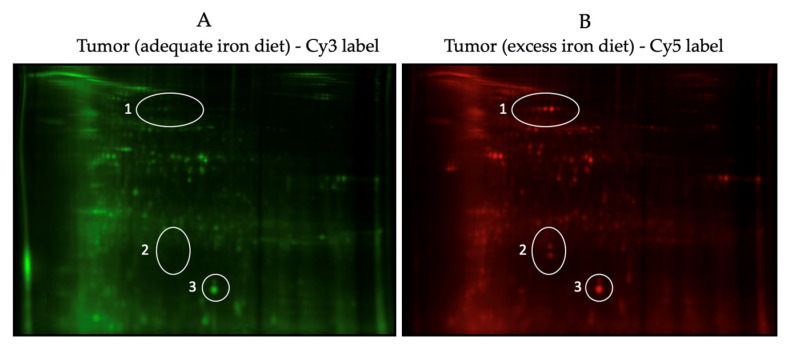
(**A**,**B**) Sample images of the 2D fluorescence difference gel electrophoresis (2D-DIGE) Cy3 and Cy5 labeled proteins from intestinal tumor tissues from adequate vs. excess iron, respectively, to illustrate differential protein expression. Circles show sample protein spots (expression) that either significantly (*p* ≤ 0.05) increased (1 and 2) or did not change (3) in tumor tissue with excess iron.

**Figure 4 nutrients-16-01316-f004:**
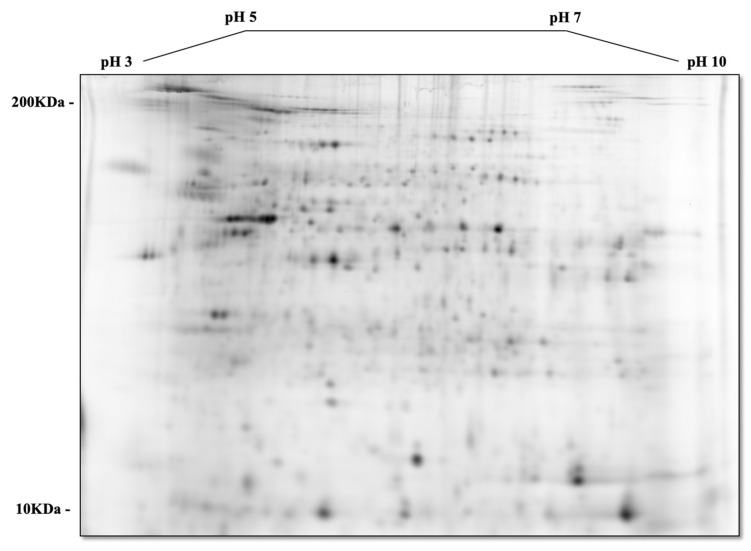
Two-dimensional (deep purple-labeled) protein spot map shown in Figure 2C showing orientation of the pH gradients on the horizontal axes, from pH 3 (left) to 10 (right), with approximate molecular mass ranges indicated along the vertical axes from 10 (near bottom) to 200 kDa (near top).

**Figure 5 nutrients-16-01316-f005:**
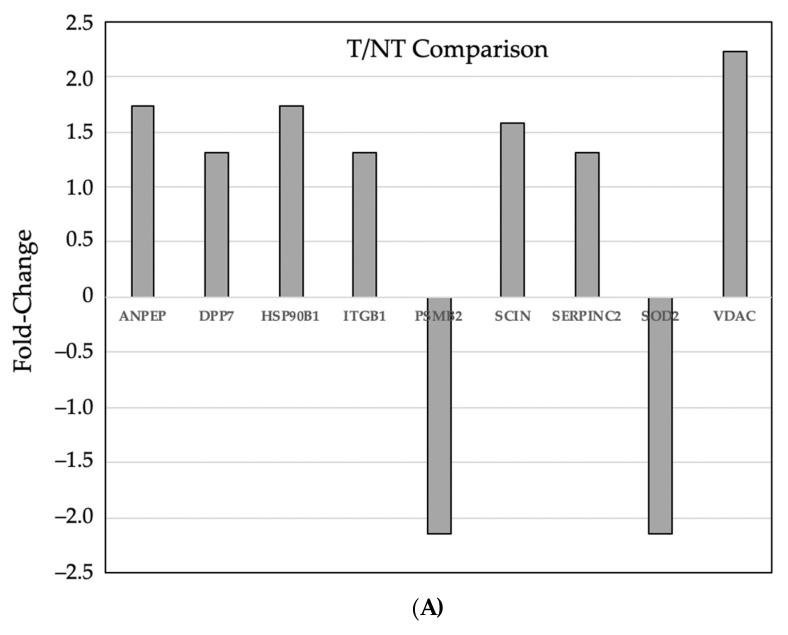
(**A**) Key differentially expressed (*p* ≤ 0.05) proteins in (**A**) tumor vs. adjacent non-tumor intestinal tissues from the adequate iron diet group and (**B**) tumor tissues from the excess vs. adequate iron diet groups. Proteins are presented by gene name. Fold-change expression differences are shown.

**Figure 6 nutrients-16-01316-f006:**
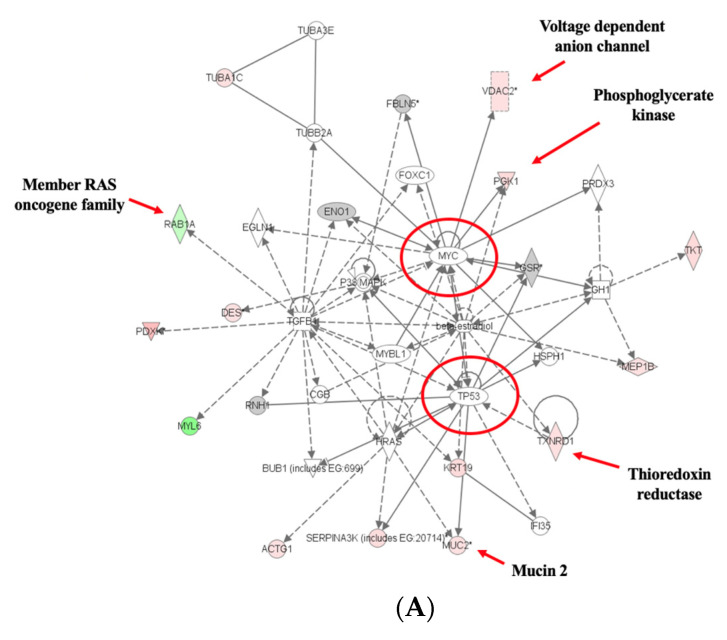
Protein networks of intestinal tumor tissues from adequate (**A**) vs. excess iron (**B**) revealed by pathway analysis. Solid lines indicate direct interaction and dashed lines indicate indirect interaction (protein abbreviations and identities are defined and described, respectively, in Table 2). An asterisk indicates that the protein is involved in cell membrane integrity or substrate transport. The intensity of red and green molecule colors indicates the degree of up-or-down-regulation, respectively. Myc- and p53- related networks (red circles) are associated with functions implemented in cancer and cell death. Select key differentially expressed protein names are shown.

**Table 1 nutrients-16-01316-t001:** Baseline diet characteristics, showing the iron (Fe)-deficient diet (AIN-93M[M]) * to which iron (ferrous sulfate monohydrate) was added to produce 45 and 450 mg Fe/kg diets **.

Formula	g/Kg	
Casein, low Cu and Fe	200.0	
DL-Methionine	3.0	
Sucrose	545.19	
Corn Starch	150.0	
Corn Oil	50.0	
Alphacel (low mineral fiber)	50.0	
Mineral Mix, Fe Deficient (81062) ***	35.0	
Vitamin Mix, AIN-76A (40077) ***	14.0	
Choline Bitartrate	2.8	
Ethoxyquin, antioxidant	0.01	
**Macronutrient**	**% dry weight**	**% kcal**
Protein	17.7	17.8
Carbohydrate	69.8	70.4
Fat	5.2	11.8

* Ref: [36]. (The base AIN-93M[M] diet was prepared by Harlan Teklad (Catalog #TD.99397; Harlan Teklad, USA)). ** Atomic absorption spectrophotometry was used to confirm iron concentrations of the diets, as previously described [37]). The mean concentration of iron assayed in the 45 and 450 mg Fe/kg diets, performed in triplicate, were 44.8 ± 0.7 mg and 446.9 ± 6.2 mg iron/kg diet, respectively. *** Diet/ingredient catalog numbers shown—all other ingredients obtained from Harlan Teklad (Harlan Teklad, USA).

**Table 2 nutrients-16-01316-t002:** Differentially expressed proteins in tumor and adjacent non-tumor intestinal tissue. Proteins presented alphabetically with gene name and GenInfo Identifier (GI ID) are those that differed (*p* ≤ 0.05) significantly between the comparisons shown. Fold-change expression differences are shown. (Key: “T/NT” = 45 mg/kg dietary iron tumor vs. non-tumor tissue; “TFe/T” = 450 mg/kg dietary iron tumor vs. 45 mg/kg dietary iron tumor tissue).

Gene	GI ID	Spot	Fold Change		Description
			T/NT	TFe/T	
ACTG1	809561	1246	1.74		actin, gamma 1
ACTN4	11230802	741		−2.55	actinin, alpha 4
ALB	26986064	1728	2.87		albumin
	33859506	1455		3.47	
ALDOB	1619606	1856		2.16	aldolase B
	15723268	1682	3.07		
ANPEP	6678664	543	1.73		alanyl (membrane) aminopeptidase
AOC3	4185817	543	1.73		amine oxidase, copper containing 3
ARG2	6753110	1698	2.75		arginase, type Il
CAPZB	83649737	1484		10.86	capping protein (actin filament) muscle -line, beta
CPS1	8393186	909	1.77		carbamoyl-phosphate synthetase 1, mitochondrial
DES	33563250	1046	1.76		desmin
DPP7	13626390	933	1.31	1.59	dipeptidyl-peptidase 7
FBN1	118197277	543	1.73		fibrillin 1
FCGBP	21410127	485		−3.83	Fc fragment of IgG binding protein
	94381948	1372	15.21		
FLNA	38257560	1728	2.87		filamin A, alpha (actin binding protein 280)
	47847514	909	1.77		
FLNB	38257404	579	1.58		filamin B, beta (actin binding protein 278)
FLNC	94377129	578	1.79	1.72	filamin C, gamma (actin binding protein 280)
	94377129	579	1.58		
GPD1	387177	1682	3.07		glycerol-3-phosphate dehydrogenase 1 (soluble)
GSR	13624751	933	1.31	1.59	glutathione reductase
GSTM3	6754086	1809		1.96	glutathione S-transferase M3
HPX	23956086	1032	2.3		hemopexin
HSP90A81	40556608	543	1.73		heat shock protein 90 kDa alpha, class B member 1
HSP90B1	6755863	543	1.73		heat shock protein 90 kDa beta (Grp94), member 1
HSPA5	1304157	741		−2.55	heat shock 70 kDa protein 5
HSPA9	42542422	741		−2.55	heat shock 70 kDa protein 8
IGH	62027409	909	1.77		immunoglobulin heavy chain complex
ITGB1	762977	933	1.31	1.59	integrin, beta 1
KHK	31982229	1468	−2.8		ketohexokinase
KRT19	6680606	1698	2.75		keratin 19
LAMC1	31791057	933	1.31	1.59	laminin, gamma 1
LCP2	31543113	741		−2.55	lymphocyte cytosolic protein 1
LCT	74192292	263		−1.95	lactase
	74192292	652	−2.86		
LGALS3BP	397800	933	1.31	1.59	lectin, galactoside-binding, soluble, 3 binding protein
MUC2	28865873	1052	1.84		mucin 2, oligomeric mucus/gel-forming
MYH12	50510675	741		−2.55	myosin, heavy chain 11, smooth muscle
PEPD	6679279	933	1.31	1.59	peptidase D
PGK1	129903	1698	2.75		phosphoglycerate kinase 1
PRDX1	6754976	1953	−2.15		peroxiredoxin 1
PRDX4	7948999	1698	2.75		peroxiredoxin 4
PSMA1	33563282	1574	3.49		proteasome subunit, alpha type, 1
	33563282	1640	2.23		
PSMA3	31981534	2155	−3.53		proteasome subunit, alpha type, 3
PSMB2	31981327	1953	−2.15		proteasome subunit, beta type, 2
PSMB4	3914439	1809		1.96	proteasome subunit, beta type, 4
RAB1A	206553	1909	−1.68		RAB1A, member RAS oncogene family
SCIN	2851563	579	1.58		scinderin
SERPINA3K	54173	861	2.39	2.21	serine peptidase inhibitor, clade A, member 3K
	54173	862	2.54	2.6	
	54173	863	2.33		
	54173	868	2.19		
SERPINB1	114158675	1296	1.58		serpin peptidase inhibitor, clade B, member 1
	114158675	1698	2.75		
SERPINB6	6678097	1246	1.74		serpin peptidase inhibitor, clade B, member 6
	6678097	1247	1.73		
SERPINC2	18252782	933	1.31	1.59	serpin peptidase inhibitor, clade C. member 1
SOD1	45597447	2137	−6.52		superoxide dismutase 1
SOD2	53450	1953	−2.15		superoxide dismutase 2
TNC	220610	543	1.75		tenascin C
	29290613	868	2.19		
TPI1	54855	1682	3.07		triosephosphate isomerase 1
TUBA1C	6678469	1032	2.3		tubulin, alpha 1c
TXNRD1	13569841	909	1.77		thioredoxin reductase 1
VDAC2	6755965	1484		10.86	voltage-dependent anion channel 2
	6755965	1640	2.23		

## Data Availability

The original contributions and data on which findings are presented in this study are included within the article and Appendix A.

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
