# Peer review of "Iron Supplementation Increases Tumor Burden and Alters Protein Expression in a Mouse Model of Human Intestinal Cancer"

_nutrients, 2024, doi:10.3390/nu16091316_

Round 1
Reviewer 1 Report
Comments and Suggestions for Authors
The experimental design is sound and the results support the hypothesis that iron supplementation accelerates intestinal tumorigenesis. The methodologies and analyses were well-executed and the data presentation is clear, making coherent conclusions that align with current scientific literature.
1. While the manuscript focuses on quantifying tumor burden in terms of area (mm^2/cm), I recommend that the authors include statistics on the number of tumors as well in the revised manuscript. In line 257, trying to plot a tumor number and burden graph would be better.
2. For table 2, visual representations might help in quickly grasping comlex data, presenting this in bar graph format would likely be more friendly for readers.
3. in line 43 to 54, consider to revise this part of introdcution.
Comments on the Quality of English Language
English is good. However, the introduction that can be optimized.
Reviewer 2 Report
Comments and Suggestions for Authors
The manuscript described the effects of excess iron on a mice model of intestinal cancer with a focus on altered protein expression. The manuscript is extremely well written and prepared. The subject is timely and interesting. The research appears to be very well performed. I only have a couple rather trivial concerns.
The manuscript mentions supplementary material that was not accessible during this review.
Add comma after "measured" in line 14.
Either report centrifuge model along with rpm or report as "x g".
